# Analysis of the Morphology and Structure of Carbon Deposit Formed on the Surface of Ni_3_Al Foils as a Result of Thermocatalytic Decomposition of Ethanol

**DOI:** 10.3390/ma14206086

**Published:** 2021-10-14

**Authors:** Pawel Jóźwik, Agata Baran, Tomasz Płociński, Daniel Dziedzic, Jakub Nawała, Malwina Liszewska, Dariusz Zasada, Zbigniew Bojar

**Affiliations:** 1Faculty of Advanced Technologies and Chemistry, Institute of Materials Science and Engineering, Military University of Technology, 00-908 Warszawa, Poland; dariusz.zasada@wat.edu.pl (D.Z.); zbigniew.bojar@wat.edu.pl (Z.B.); 2Faculty of Materials Science and Engineering, Warsaw University of Technology, 02-507 Warszawa, Poland; tomasz.plocinski@pw.edu.pl; 3Faculty of Advanced Technologies and Chemistry, Institute of Chemistry, Military University of Technology, 00-908 Warszawa, Poland; daniel.dziedzic@wat.edu.pl (D.D.); jakub.nawala@wat.edu.pl (J.N.); 4Institute of Optoelectronics, Military University of Technology, 00-908 Warszawa, Poland; malwina.liszewska@wat.edu.pl

**Keywords:** CNT/CNF morphology, Ni_3_Al foils, ethanol decomposition

## Abstract

This article presents the results of investigations of the morphology and structure of carbon deposit formed as a result of ethanol decomposition at 500 °C, 600 °C, and 700 °C without water vapour and with water vapour (0.35 and 1.1% by volume). scanning electron microscopy (SEM) and scanning transmission electron microscopy (STEM) observations as well as energy dispersive X-ray spectrometry (EDS), X-ray diffraction (XRD), and Raman spectroscopic analyses allowed for a comprehensive characterization of the morphology and structure of cylindrical carbon nanostructures present on the surface of the Ni_3_Al catalyst. Depending on the reaction mixture composition (i.e., water vapour content) and decomposition temperature, various carbon nanotubes/carbon nanofibres (CNTs/CNFs) were observed: multiwalled carbon nanotubes, herringbone-type multiwall carbon nanotubes, cylindrical carbon nanofibers, platelet carbon nanofibers, and helical carbon nanotubes/nanofibres. The discussed carbon nanostructures exhibited nickel nanoparticles at the ends and in the middle part of the carbon nanostructures as catalytically active centres for efficient ethanol decomposition.

## 1. Introduction

Carbon nanotubes (CNTs) and carbon nanofibres (CNFs) exhibit unique physical and chemical properties compared to classical materials. These properties include high strength, low density, and high thermal and electrical conductivity. Current and future applications of these materials include: high-performance composites, electrochemical devices, hydrogen storage, sensors, probes, coatings, and films [1,2,3,4,5]. The most popular methods for producing CNTs/CNFs are arc discharge (AD), pulse laser deposition (PLD), and chemical vapour deposition (CVD). Moreover, CVD is modified by combining with heterogeneous catalysis, i.e., catalytic chemical vapour deposition (CCVD) [2,3,5,6]. The CCVD method has gained considerable attention due to its low cost and relatively easy scalability. The catalytic function in the CCVD method is carried out mainly by transition metals (i.e., nickel, iron, and cobalt). However, catalysts based on abovementioned metals tend to have limited control over both the dimension and structure of carbon nanostructures.

It is worth noting that the properties of CNT/CNF, apart from their morphology, largely depend on their structure, including the number and spatial orientation of the layers of carbon atoms that form them (in some works, these layers of carbon atoms are equated with graphene layers and, in others, with base planes (002) in the crystalline structure of graphite). For example, the highest strength among carbon nanostructures is demonstrated by cylindrical carbon nanotubes formed either by a parallel (rolled around the axis of the nanotube) graphene layer (this is the case of single-walled carbon nanotube-SWCNT) or by a pack of mutually parallel, rolled around the axis of the nanotube, layers of carbon atoms treated in this case as a package of (spaced apart) graphene layers or as a package of base planes in the graphite structure (this is the case of multiwalled carbon nanotube-MWCNT). Changing the spatial arrangement of the discussed layers of carbon atoms to perpendicular to the axis of the carbon nanostructure causes a change in the morphological form of this nanostructure (this is the case of platelet carbon nanofibers, p-CNF). Although this type of carbon nanostructure has a significantly limited strength, it is also characterized by a high ability to adsorb foreign atoms, which is of great importance, e.g., in the field of hydrogen storage. Strict control of the CCVD process with the use of a properly selected catalyst can therefore ensure the possibility of producing the assumed type of CNT/CNF of an acceptable quality (for a specific application).

In this study, Ni_3_Al intermetallic phase alloy foils were used as a catalyst in the CCVD method. It has been previously confirmed [7,8,9,10,11,12] that Ni_3_Al has high catalytic activity (greater than Ni) in the following processes: decomposition of hydrocarbons (methane, cyclohexane), as well as methyl alcohol for “hydrogen production,” or air purification from chemical warfare agents. As far as we know, the available literature lacks research on the thermocatalytic decomposition of ethanol (EtOH) using the Ni_3_Al catalyst in the form of a foil or powder and on the synthesis of carbon nanostructures as a product of decomposition of this alcohol.

The main purpose of this work was to investigate the temperature and water vapour presence effect on the structure and morphology of CNT/CNF-type carbon deposits formed on the surface of Ni_3_Al foil/catalyst as a result of the thermocatalytic decomposition of ethanol. Due to its availability (it is made from plant products such as corn, sugarcane, cereals, sugar beet, etc.) and its limited toxicity, ethanol is a widely used renewable fuel. It is also popular in processes aimed at the production of CNT/CNF-type nanostructures [3,13,14,15]. According to the assumption, the addition of water vapor to the reaction mixture with ethanol should result in a significant improvement in the quality (assessed by the repeatability of geometric and structural parameters) of the obtained carbon nanostructures, which will be shown in this article.

## 2. Materials and Methods

The carbon deposition studies presented in this article are concerned with the solid thermocatalytic reaction products of ethanol decomposition formed on the surface of Ni_3_Al foils. Three compositions of the reaction mixture with different water vapour contents and similar EtOH contents were used in this study (Table 1). For ethanol decomposition, foils made of an alloy based on the Ni_3_Al intermetallic phase matrix (Ni-19Al-0.17Zr-0.15B at.%) were used. They were obtained by multivariate thermoplastic processing, and details of the fabrication procedure were presented in previous articles [16,17]. Before thermocatalytic testing, Ni_3_Al foils were mechanically ground using 80-grit SiC paper, cut into 10 mm × 3 mm pieces, and defatted in an ultrasonic cleaner using acetone. Then, the prepared material was placed in a quartz fixed-bed type reactor. The processes of ethanol decomposition were performed isothermally using the following experimental conditions: catalyst mass–0.8 g; time–3 h; temperature–500 °C, 600 °C, and 700 °C; with argon (99.999% Ar) as the carrier gas (5 L/h flow rate).

A Quanta 3D FEG scanning electron microscope coupled with an EDS/EDAX spectrometer and a Hitachi HD 2700 scanning transmission electron microscope were used to describe the changes in the carbon deposit morphology formed on the Ni_3_Al foil surface over time during the thermocatalytic tests. Before STEM examination, the deposit was mechanically scraped from the surface and placed on amorphous carbon film (AGAR S147H). In the high-resolution transmission electron microscopy (HRTEM) analysis, DigitalMicrograph software with additional scripts was used [18]. The inverse fast Fourier transform (IFFT) was used to determine the structural details of the observed nanostructures. The values of the interplanar distances (d_hkl_) presented in the pictures constitute an average value measured using 7–10 particular planes. Additionally, the surface of the Ni_3_Al film with carbon deposit was observed using the Keyence VHX 950F light microscope.

After catalytic tests, X-ray phase analysis of the Ni_3_Al surface was performed using an ULTIMA IV diffractometer equipped with a Co lamp using a 40-kV voltage and a 40-mA current. In addition, due to a large amount of deposit present on the surface of the Ni_3_Al foils in mixture #1, XRD studies of the deposition itself were also performed after its mechanical removal from the foil surface.

Analysis of the defect density and the graphitization degree of the produced carbon deposit was performed using a Renishaw InVia Raman microscope (Renishaw plc., Wotton-under-Edge, UK) equipped with an EMCCD detector (Andor Technology Ltd., Oxford Instruments, Belfast, UK). The Raman signal was acquired using laser radiation with a 532 nm wavelength. The laser beam was directed to the sample through a 20× (N.A. = 0.40) objective lens. The wavelength of the instrument was calibrated using an internal silicon wafer, and the spectrum was centred at 520.5 cm^−1^. Raman measurements were performed in the range from 100 cm^−1^ to 3200 cm^−1^.

## 3. Results

### 3.1. SEM and EDS/SEM Examinations

The initially clean and shiny surface of Ni_3_Al foils, as a result of the thermocatalytic decomposition of ethanol, was covered with a macroscopically visible layer of carbon deposit, which was visibly reduced with increasing water vapour content in the reaction mixture (Figure 1). Thus, the highest presence of solid reaction products on the surface of Ni_3_Al foils was observed after the process performed with the reaction mixture containing ethanol and argon as a carrier gas (i.e., mixture #1). The lowest presence was obtained for the reaction mixture with the highest content of water vapour (i.e., mixture #3—Table 1).

Regardless of the steam contribution and the decomposition process temperature, the surface of Ni_3_Al foils after thermocatalytic ethanol decomposition showed a predominant presence of CNT/CNF-type tubular carbon nanostructures. Nevertheless, their diameter depended on the process temperature and steam contribution (compare Figure 2, Figure 3 and Figure 4). The ethanol decomposition process performed without steam (i.e., mixture #1) at 500 °C resulted in the presence of CNTs/CNFs with a dominant diameter (D) of approximately 150 nm. At the same time, the presence of much thinner nanostructures showing D in the vicinity of 50 nm was noticed (Figure 2a,b).

Increasing the process temperature resulted in a gradual decrease in the size of the observed nanostructures. The decomposition realized at 600 °C led to the formation of CNTs/CNFs exhibiting a dominant D value of ~50 nm, with only the local presence of larger diameters of D ≈ 100 nm (Figure 2c,d). In contrast, decomposition at 700 °C resulted in a further reduction in the diameter of the observed tubular nodular nanostructures to a dominant D value of ~30 nm, with a local presence of systems with a slightly larger diameter, i.e., 50 nm (Figure 2e,f). Of note, metallic nanoparticles located mainly at the ends of nanostructures were observed. The localized presence of metallic nanoparticles was also observed in the depositions formed after ethanol decomposition at 500 °C in the central part of CNTs/CNFs resembling two cones connected by bases (see enlargement in Figure 2a). The mentioned nickel nanoparticles are effective catalytic active centres that form in the surface layer of Ni_3_Al alloys during the thermocatalytic decomposition of many chemical compounds such as methanol [7,8], methane [19,20], and cyclohexane [21]. According to previous research by T. Hirano [9] and D. Chun et al. [10], the presence of Ni nanoparticles is a consequence of selective oxidation and hydroxylation processes of aluminium from the surface layer of Ni_3_Al foils.

The presence of water vapour in the reaction mixture (mixtures #2 and #3—see Table 1) noticeably decreased the diameter of the cylindrical carbon nanostructures while increasing their size uniformity (compare Figure 2, Figure 3 and Figure 4). The decomposition of ethanol with 0.35 vol.% H_2_O (mixture #2) performed at 500 °C resulted in the presence of CNT/CNF nanostructures on the surface of Ni_3_Al foils with a dominant diameter of ≈70 nm, at 600 °C – ≈ 50 nm, and at 700 °C even up to 30 nm (Figure 3). Of note, the process performed for this composition of the reaction mixture at the lowest temperature (i.e., 500 °C) also led to the local formation of helical carbon nanostructures on the surface of Ni_3_Al foils (Figure 3b).

An increase in water vapour content (1.1 vol.% H_2_O—mixture #3) resulted in a further, visible, and even macroscopic reduction in carbon deposition on the surface of Ni_3_Al foils. A further decrease in the diameter of CNT/CNF structures was observed on the surface of the Ni_3_Al catalyst, with a simultaneous increase in the homogeneity of their size (Figure 4). The carbon deposit formed due to the ethanol decomposition process performed at 500 °C showed the dominant diameter of tubular nanostructures D ≈ 60 nm. An increase in process temperature, similarly to the above-discussed reaction mixtures (without water vapour content-mixture #1 and with its 0.35 vol.%) (Figure 2 and Figure 3 ), resulted in a further reduction in CNT/CNF diameter; at 600 °C, the dominant diameter was approximately 40 nm and approximately 20 nm at 700 °C.

Investigations of the chemical composition of carbon deposits formed on the surface of Ni_3_Al thin foils after thermocatalytic decomposition of ethanol (mixtures #1–#3) using energy dispersive X-ray spectrometry (EDS/SEM) allowed to identify its key elements (Figure 5, Figure 6 and Figure 7). The dominant carbon content in the EDS analysed deposit, visible in the linear distribution of the main constituent elements, confirmed the morphologically described presence of the above-mentioned CNT/CNF. On the other hand, metallic nanoparticles common at the ends of cylindrical carbon nanostructures, as well as quite numerous in individual fragments of their length, showed significant enrichment in nickel, regardless of the H_2_O content in the reaction mixture. At the same time, the content of other elements in the micro-volume of metallic particles present in the carbon deposit was strongly reduced.

In contrast, this study showed no significant differences in the chemical composition of the analysed areas (Figure 5, Figure 6 and Figure 7) of the deposit formed on the surface of Ni_3_Al catalyst as a result of ethanol decomposition with or without water vapour (mixtures #1–#3).

### 3.2. STEM Analysis

STEM research allowed, apart from confirming the morphological features of the observed CNT/CNFs, also a detailed observation and analysis of their substructure. Observations of the deposit were limited to the boundary states, i.e., those obtained after decomposition of ethanol without the presence of water vapour (mixture #1) and with the highest water vapour content (mixture #3) performed at 500 and 700 °C (Figure 8, Figure 9, Figure 10 and Figure 11).

The carbon deposit formed on the surface of Ni_3_Al catalyst by decomposition at 500 °C of ethanol without steam (i.e., mixture #1) showed a predominant presence of two nanofibre types, i.e., cylindrical carbon nanofibres (c-CNFs) with an average diameter D of approximately 150 nm and locally multiwalled carbon nanotubes (MWCNTs) with an average diameter D ≈ 50 nm [5,12,22].

HRTEM observations using the fast inverse Fourier transform enabled the visualization of layered arrangements of carbon atoms analogous to the crystallographic planes (002) in the graphite structure. Carbon atoms’ planes of type (002) oriented along the CNF/CNT axis (Figure 8d,e) were identified, and their mutual distances were determined. Smaller values were indicated for MWCNTs (d_002_ = 0.34 nm) than for CNFs (d_002_ = 0.35 nm) (Figure 9c,d). The difference in the distance values between the type (002) planes was associated with the greater presence of impurities and/or defects both in the structure and in the mutual arrangement of the graphite-like sheets forming c-CNFs.

The presence of nickel in nanoparticles localized both at the ends and inside the CNF/CNT nanostructures, as observed by EDS/SEM, was also confirmed by EDS/STEM studies (not presented in this work) and high-resolution images of the structure obtained using IFFT (Figure 8d–f). The measured values of interplane distances d_hkl_ = 0.20 nm (Figure 8e,f) were assigned to the type (111) nickel face-centred cubic (fcc) plane (according to ICCD PDF 4+ card no. 00–004-0850 d_Ni(111)_ = 0.2034 nm). However, due to the lack of a diffraction image with a larger representation of the crystal structure, we cannot exclude the presence of nickel with a hexagonal structure for the (101) plane d_Ni(101)_ = 0.2033 nm (ICCD PDF 4+ card no. 00–002-8298). One should pay attention to nanoparticles observed by SEM (Figure 2a) in the central part of carbon nanostructures, resembling two cones connected by bases. They exhibited a distinctive area in the “junction” zone relative to the rest of the particle, showing interplanar distances of d_hkl_ = 0.22 nm. This may be a consequence of the presence of nickel with a hexagonal structure, as claimed by Chun et al. [10], for which the plane of type (002) d_002_ is equal to 0.2160 nm (ICCD PDF 4+ card no. 00–002-8298).

An increase in the decomposition temperature of ethanol present in mixture #1 (without H_2_O) up to 700 °C resulted in a marked decrease in the diameter of the cylindrical carbon nanostructures. At the same time, an increase in the proportion of multiwalled carbon nanotubes, with an average diameter of D ≈ 50 nm, was observed (Figure 9). Examination of the carbon deposition formed on the surface of Ni_3_Al foils by STEM techniques revealed the localization of Ni nanoparticles at the tip and inside the carbon nanostructures (Figure 9a,b). Similar observations were noted for the process performed at 500 °C.

The HRTEM images of the deposit allowed for the separation of individual layers of carbon atoms, identical to the graphite crystallographic plane (002), the interplanar distances of which were determined using the inverse fast Fourier transform. These values were d_002_ = 0.34 nm and d_002_ = 0.35 nm for MWCNTs and CNFs, respectively (Figure 9c,d).

The Ni nanoparticles present both along the length (inside) and at the ends of the CNTs showed the crystal structure visible in the HRTEM images. The interplanar distances of d_hkl_ = 0.20 nm corresponded in this case to the (111) nickel face-centred cubic (fcc)-type plane (d_Nifcc(111)_ = 0.2034 nm). As in the case of the analysis of the deposition after ethanol decomposition at 500 °C presented above, we cannot exclude the presence of nickel with a hexagonal structure showing a similar value for the type (101) plane, i.e., d_Nihex(101)_ = 0.2033 nm (Figure 9d).

The introduction of water vapour into the reaction mixture resulted in a significant reduction in the diameter of the cylindrical carbon nanostructures observed in STEM studies (compare Figure 8a,b, Figure 9a,b, Figure 10a,b, and Figure 11a). Investigations of the deposit formed as a result of decomposition of ethanol with 1.1 vol.% H_2_O by STEM showed the dominant presence of multiwalled carbon nanotubes with an average diameter of approximately 60 nm. At the same time, the presence of significantly smaller D ≈ 15 nm was also observed. The distance between the carbon atoms sheets forming them was d_002_ = 0.34 nm (Figure 10d).

Platelet carbon nanofibers (p-CNFs) consisting of carbon atom layers arranged perpendicular to the fibre axis with interplanar distances d_002_ = 0.35 nm were also observed locally (Figure 10c).

The Ni nanoparticles present at the ends and in the middle part of MWCNTs had a crystalline structure showing interplane distances of d_hkl_ = 0.18 nm (Figure 10d). It was assigned to the plane type (200) of face-centred cubic (fcc) nickel, for which d_(200)_ was 0.1762 nm, according to ICCD PDF 4+ sheet no. 00–004-0850.

Increasing the temperature of decomposition of ethanol with 1.1 vol.% water vapour (mixture #3) resulted in a further reduction in the diameter of the nanostructures in the carbon deposit. The deposit was dominated by multiwalled carbon nanotubes (MWCNTs) with a diameter of D ≈ 20 nm (Figure 11a). In the case of MWCNTs, distances between carbon atom layers of (002) type, d_002_ = 0.34 nm were observed (Figure 11e). Locally, multi-walled carbon nanotubes of the herringbone type (h-MWCNTs), composed of layers of carbon atoms with interplanar distances identical to the graphite base plane (002), in this case equal to d_002_ = 0.35 nm, arranged at an acute angle to the nanostructure axis, were also observed (Figure 10b and Figure 11d). Similar to the cases discussed above, Ni nanoparticles were present at the ends and inside MWCNTs (Figure 11b,c). They exhibited a crystal structure with interplanar distances of d_hkl_ = 0.18 nm (Figure 11f). This was attributed to the type (200) nickel face-centred cubic (fcc) plane for which d_(200)_ = 0.1762 nm according to ICCD PDF 4+ sheet no. 00–004-0850.

Notably, the deposit obtained for this state is characterized by a distinctly higher arrangement of the parallel carbon atom layers forming them (compare Figure 8, Figure 9, Figure 10 and Figure 11f).

### 3.3. XRD Examinations

X-ray phase analysis of the carbon deposit formed on the surface of Ni_3_Al foils due to the thermocatalytic decomposition of ethanol with (mixture #2 and mixture #3) or without water vapour (mixture #1) corresponds to the presented SEM and STEM electron microscopic observations. All obtained diffractograms show reflections belonging to the Ni_3_Al phase (as a substrate), the nickel phase, and a different proportion of the (002) reflection characteristic for the basic plane in graphite structure (Figure 12 and Figure 13). The surface of Ni_3_Al foils after decomposition of ethanol (without steam) (mixture #1) and with its lowest content, i.e., 0.35 vol.% (mixture #2), showed on the diffractograms, independent of the process temperature, the presence of reflection coming from the graphite plane (002), which is typical for CNT systems (Figure 12a, FigureFigure 13a). The presence of Ni nanoparticles in the deposit, indicated in the discussion of electron microscopy studies (SEM, STEM) and chemical composition analysis, was also confirmed by the presence of reflections coming from the (111), (200), (220), and (311) planes. In contrast, reflections from the graphite plane (002) are not visible in the diffractogram derived from the surface of the Ni_3_Al foils after ethanol decomposition at 1.1 vol.% water vapour. This phenomenon is related to the already mentioned smallest deposit contribution in comparison to the Ni_3_Al substrate.

In addition, direct observations were made based on the material state after ethanol decomposition without steam (mixture #1). It showed the highest presence of carbon deposition on the surface of Ni_3_Al foils (after substrate removal) (Figure 12b). This form of the test material allowed both to eliminate the “background” from the substrate (i.e., the Ni_3_Al phase). Additionally, the volume of the carbon deposit interacting with the X-ray beam was increased, and the intensity of the observed spectrum was improved. Apart from the abovementioned plane (002), the diffractogram contains other less intense reflections from planes of type (hk0). They are a consequence of a “honeycomb” lattice in both the single graphene layer and other graphite-like structures.

In the discussion of reflections from carbon structures, one should also consider the presence of carbon with a turbostratic structure (2D graphite). It corresponds to graphene-like layers stacked (one above the other) with random rotations or translations without preserving the ABAB… sequence typical for graphite. As a consequence, it only shows the presence of reflexion from parallel layers of carbon atoms corresponding to planes of type (002) in graphite structure (Figure 12b) [5,12,23,24].

### 3.4. Raman Spectroscopy Examinations

Both the presence of water vapour in the reaction mixture and the temperature of the ethanol decomposition process significantly affected the quality and degree of graphitization of the observed carbon nanostructures evaluated by Raman spectroscopy. All obtained spectra showed the presence of three main bands with intensities dependent on the decomposition conditions used (Figure 14):

-D (disorder band)—corresponding to the frequency of ~1350 cm^−1^—comes from bond vibrations of carbon atoms occurring in sp^3^ hybridization. This indicates the presence of defects and disorders in the structure of CNTs,-G (graphitic band)—corresponding to the frequency of ~1580 cm^−1^—which is the degree of graphitization, indicates the order and purity of the CNT structure;-G’ (2D)—occurs at approximately twice the D frequency, i.e., ~ 2700 cm^−1^, indicating the presence of stresses in the CNT structure.

The highest intensity of band D was observed, irrespective of the composition of the reaction mixture (i.e., the proportion of water vapour) at the lowest decomposition temperature—500 °C. With increasing temperature, there was an apparent weakening of this band (Figure 14). In agreement with the STEM observations (Figure 8, Figure 9, Figure 10 and Figure 11) indicating the presence of MWCNT/CNF-type nanostructures, the obtained Raman spectra (Figure 14) do not show the presence of radial breathing mode (RBM) band characteristic of single-walled nanotubes (i.e., D < 2 nm). It occurs at frequencies lower than 200 cm^−1^ [25,26,27]. To determine the quality and degree of graphitization observed on the MWCNT/CNF surface, the I_D_/I_G_ parameter was determined, which was calculated from the maximum peak intensity. This provides a measure of the presence of defects in the investigated carbon nanostructures as well as the distortion of the arrangement of layers of carbon atoms, identical to the crystallographic plane (002) of graphite observed in STEM studies. A broader discussion was conducted on the STEM analysis of the deposition morphology (Figure 8, Figure 9, Figure 10 and Figure 11). The ordering of the carbon deposit formed on the surface of Ni_3_Al foils clearly improved by increasing both the presence of water vapour and the decomposition temperature (Figure 14). The highest quality CNT/CNF nanostructures (i.e., the lowest I_D_/I_G_ parameter value) were observed after ethanol decomposition for the highest water vapour content in the reaction mixture (i.e., 1.1 vol.%). For this sample at 700 °C, the I_D_/I_G_ parameter was 0.75.

## 4. Discussion

The obtained results indicate a significant influence of the analysed factors (water vapour content and decomposition temperature) on the morphology of the carbon deposit formed on the Ni_3_Al foil surface (Figure 2, Figure 3, Figure 4, Figure 5, Figure 6, Figure 7, Figure 8, Figure 9, Figure 10, Figure 11, Figure 12, Figure 13 and Figure 14). By changing their values on the surface of Ni_3_Al foils, carbon nanostructures were observed in the form of:

(a)Multiwalled carbon nanotubes MWCNTs composed of parallel carbon atom layers (identical to the crystallographic plane (002) of graphite arranged parallel to the axis) (Figure 8d, Figure 9c, Figure 10d, and Figure 11e,f), which were present in all investigated material states;

MWCNTs formed after decomposition of ethanol without steam (mixture #1) at 500 °C showed the following distances between adjacent planes (002) of graphite: d_002_ = 0.35 nm (Figure 8d) or d_002_ = 0.34 nm after decomposition of this reaction mixture realized at 700 °C (Figure 9c). In contrast, for MWCNTs formed as a result of the process performed with 1.1 vol.% steam (mixture #3), regardless of the decomposition temperature, d_002_ = 0.34 nm (Figure 10d and Figure 11e);

(b)cylindrical carbon nanofibres (c-CNFs) are built of parallel carbon atom layers (identical to the crystallographic plane (002) of graphite arranged parallel to their axes, also filling the core of these nanostructures) (Figure 8c and Figure 9d—present after decomposition of EtOH without water vapour (mixture #1) at 500 °C and 700 °C (locally));

After decomposition at 500 °C, cylindrical carbon nanofibres exhibited distances of d_002_ = 0.35 nm and d_002_ = 0.34 nm after decomposition at 700 °C (Figure 9d);

(c)platelet carbon nanofibre (p-CNFs) presented locally only after decomposition at 500 °C of ethanol with 1.1 vol.% steam (mixture #3) having d_002_ = 0.35 nm (Figure 10c);(d)herringbone-type multiwalled carbon nanotubes (h-MWCNTs) presented locally only after decomposition of ethanol with 1.1 vol.% steam (mixture #3) at 700 °C, showing d_002_ = 0.35 nm (Figure 11d);(e)helical carbon nanotubes/nanofibres presented locally only after decomposition of ethanol with 0.35 vol.% steam (mixture #2) at 500 °C (Figure 3b), visible only in SEM studies.

The indicated differences in distances between layers of carbon atoms, identical to the crystallographic plane (002) of graphite (i.e., d_002_ = 0.34 nm or 0.35 nm) are a consequence of the presence of defects in their mutual arrangement and/or the presence of impurities. The limited quality of the CNT/CNF nanostructures formed on the surface was also observed in HRTEM studies. Among others, striations constituting local domains with similar orientations were observed concerning electron microscope beam (Figure 8d, Figure 9c,d, and Figure 11d) and by Raman spectroscopy (Figure 14). The highest arrangement of carbon deposit was observed after ethanol decomposition, with the highest proportion of water vapour in the reaction mixture (1.1 vol.%) at 700 °C. This sample’s I_D_/I_G_ parameter reached the lowest value, i.e., 0.75 (Figure 11 and Figure 14 ).

The observed effect of the temperature of the decomposition process on the morphology of carbon deposit (Figure 2, Figure 3 and Figure 4 and Figure 8, Figure 9, Figure 10 and Figure 11) in the literature [6,28,29] is considered an essential factor affecting the diameter of CNTs/CNFs and limiting their defects. A dominant presence of cylindrical carbon nanofibres was observed on the surface of Ni_3_Al foils after decomposition of ethanol without water vapour (mixture #1) at 500 °C (Figure 8). This is a consequence of the difference between the diffusion rate of carbon atoms in the catalyst particle and the nucleation rate [30]. When both diffusion and nucleation rates are relatively low, the efficiency of diffusive transport of carbon atoms dominates over nucleation efficiency. This phenomenon was possible due to the relatively large contribution of surface diffusion. As a result, parallel carbon atom layers were observed to form across the metal/carbon interface. These can be seen as cylindrical carbon fibres with a filled core (Figure 8a–c). At relatively high temperatures, when the “reception” of carbon atoms by nucleating carbon atom layers is much faster than the possibility of diffusive transport of these atoms, there is no driving force for the nucleation of a single, graphene-like layers in remote zones with a longer diffusion path (i.e., in the core region of the potential fibre). Consequently, carbon nanotubes exhibiting an unfilled core are dominant at 700 °C (Figure 9a–c, Figure 10d, and Figure 11d–f).

The presence of water vapour in the reaction mixture favours, as proven before [31,32], the quality of the obtained carbon nanostructures obtained by binding “excess” amorphous carbon to CO. As a consequence, the CNT/CNF-type carbon nanostructures formed on the surface of Ni_3_Al foils with 1.1 vol.% water vapour are characterized by a significantly higher identity observed in both STEM observations (Figure 8, Figure 9, Figure 10 and Figure 11) and Raman spectroscopy studies (Figure 14).

Regardless of the proportion of water vapour in the reaction mixture, metallic nanoparticles were observed at the ends and in the middle part of the CNT/CNF. They exhibited a dominant presence of nickel in EDS studies (Figure 5, Figure 6 and Figure 7). Analysis of HRTEM images of these areas allowed us to determine the distance of the lattice plane—d_hkl_ = 0.20 nm and d_hkl_ = 0.18 nm (Figure 8e–f, Figure 9d, Figure 10d, and Figure 11f). They were assigned to the A1 structure nickel planes (111) and (200), respectively. This Ni allotropic variant was also observed in XRD studies (Figure 12 and Figure 13). Such identification has also been confirmed by many authors: Hirano [9], Chun et al. [10], Jang et al. [33], and Xu [19]. However, other studies show the presence of Ni_3_C carbide, which is not treated unambiguously in the available literature. In some works, it is questioned or treated as a symptom of catalyst deactivation [34,35]. In contrast, in other studies, authors point to its important role in the initial decomposition period, during which it decomposes into carbon and nickel [6,36]. However, it is worth mentioning the limited stability of this phase, which is significantly affected by the conditions of the implemented process [36,37]. Therefore, the presence of secondary reactions (that may significantly influence the final structure of Ni particles studied ex situ) occurring after the decomposition process cannot be excluded.

Decomposition of ethanol without steam (mixture #1) at 500 °C resulted in the presence of Ni particles in the middle part of CNF in the form of two base-jointed cones (Figure 2a and Figure 8a). HRTEM studies showed the existence of a distinctive region with interplane distances d_hkl_ = 0.22 nm in the “cone junction” zone. This may be a consequence of the presence of hexagonal nickel, which for plane (002) shows, according to ICCD PDF 4+ 00–002-8298, the value of d_002_ = 0.2160 nm. Such identification, despite the limited information on the crystallographic structure of this area, is confirmed by literature data [10,38,39]. The studies proved the presence of (in similar biconical Ni nanoparticles with wall-centred regular structure) hexagonal nickel or its similar crystalline structure disordered Ni_3_C carbide. Those structures were observed only in the central part (i.e., the area of connection of the cone bases). The absence of the indicated nickel allotropic variety (Figure 12) in the diffractogram was associated with its limited contribution to the deposition present on the Ni_3_Al foil surface.

## 5. Conclusions

Based on the obtained results of the conducted research, the following final conclusions were made:(a)Ni_3_Al foils are effective catalysts in the CCVD process for the formation of CNT/CNF-type carbon nanostructures as a result of ethanol decomposition.(b)Thermocatalytic decomposition of ethanol results, depending on the process temperature and the proportion of water vapour, in the formation of a deposit with a dominant presence of multiwalled carbon nanotubes or cylindrical carbon nanofibres or locally present herringbone-type multiwalled carbon nanotubes, platelet carbon nanofibres, and helical carbon nanotubes/nanofibres.(c)The introduction of 1.1 vol.% water vapour into the reaction mixture at 700 °C results in the formation of multiwalled carbon nanotubes with a diameter of approximately 20 nm, showing high ordering.(d)The presence of Ni nanoparticles involved in the ethanol decomposition process and the formation of carbon nanostructures were observed in the CNT/CNF structure irrespective of the steam content and decomposition temperature.

## Figures and Tables

**Figure 1 materials-14-06086-f001:**
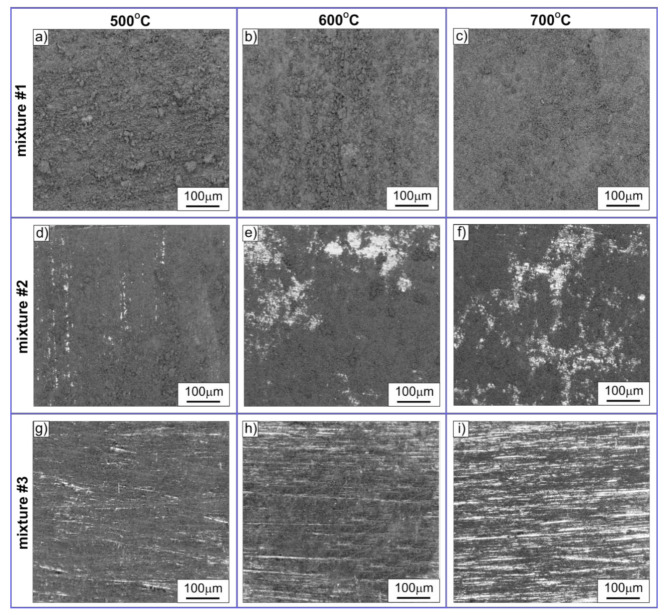
LM morphology of Ni_3_Al foils surface layer after decomposition of: (**a**–**c**) mixture #1, (**d**–**f**) mixture #2, (**g**–**i**) mixture #3 at temperature: (**a**,**d**,**g**) 500 °C, (**b**,**e**,**h**) 600 °C, and (**c**,**f**,**i**) 700 °C.

**Figure 2 materials-14-06086-f002:**
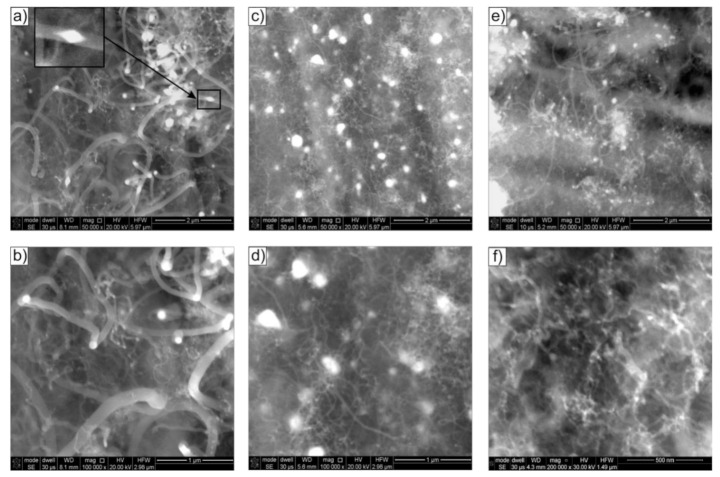
SEM morphology of Ni_3_Al foils surface after ethanol decomposition without water steam addition (mixture #1) at: (**a**,**b**) 500 °C, (**c**,**d**) 600 °C, and (**e**,**f**) 700 °C.

**Figure 3 materials-14-06086-f003:**
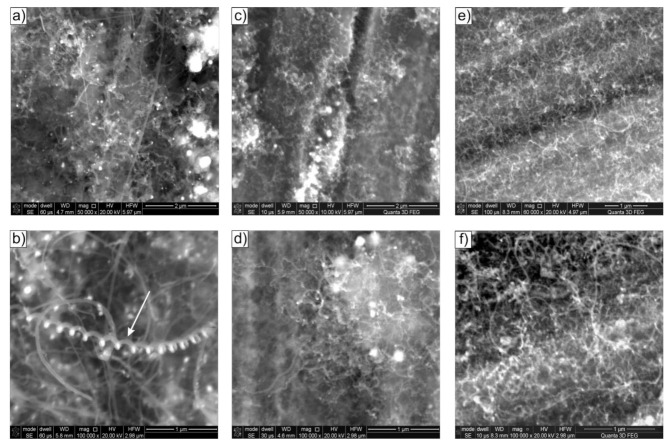
SEM morphology of Ni_3_Al foils’ surface after decomposition of ethanol with addition of 0.35 vol.%. water steam (mixture #2) at: (**a**,**b**) 500 °C, (**c**,**d**) 600 °C, and (**e**,**f**) 700 °C.

**Figure 4 materials-14-06086-f004:**
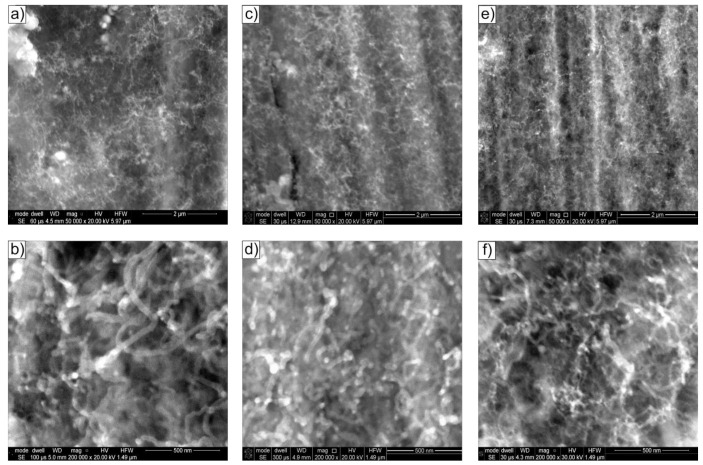
SEM morphology of Ni_3_Al foils surface after decomposition of ethanol with addition of 0.5 wt.% water steam (mixture #3) at: (**a**,**b**) 500 °C, (**c**,**d**) 600 °C, and (**e**,**f**) 700 °C.

**Figure 5 materials-14-06086-f005:**
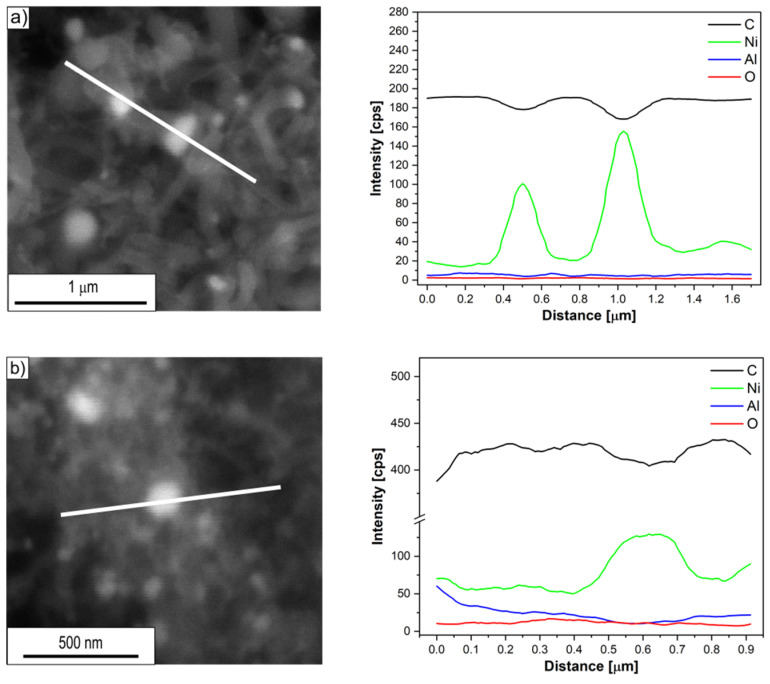
Linear EDS/SEM analysis of the CNT/CNF area with nickel-like nanoparticles formed after decomposition of ethanol without water vapour addition (mixture #1) at: (**a**) 500 and (**b**) 700 °C.

**Figure 6 materials-14-06086-f006:**
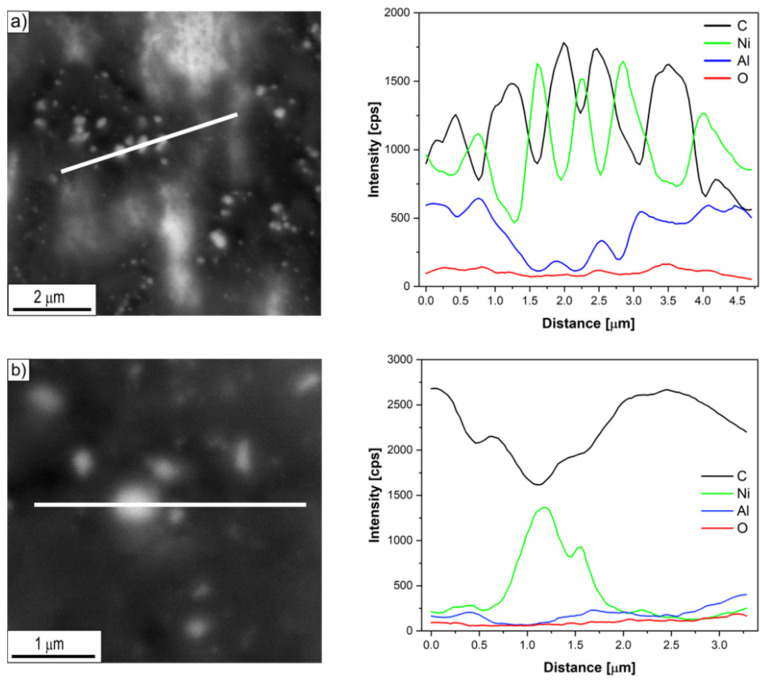
Linear EDS/SEM analysis of the CNT/CNF area with nickel-like nanoparticles formed after decomposition of ethanol with 0.35 vol.% addition of water vapour (mixture #2) at: (**a**) 500 and (**b**) 700 °C.

**Figure 7 materials-14-06086-f007:**
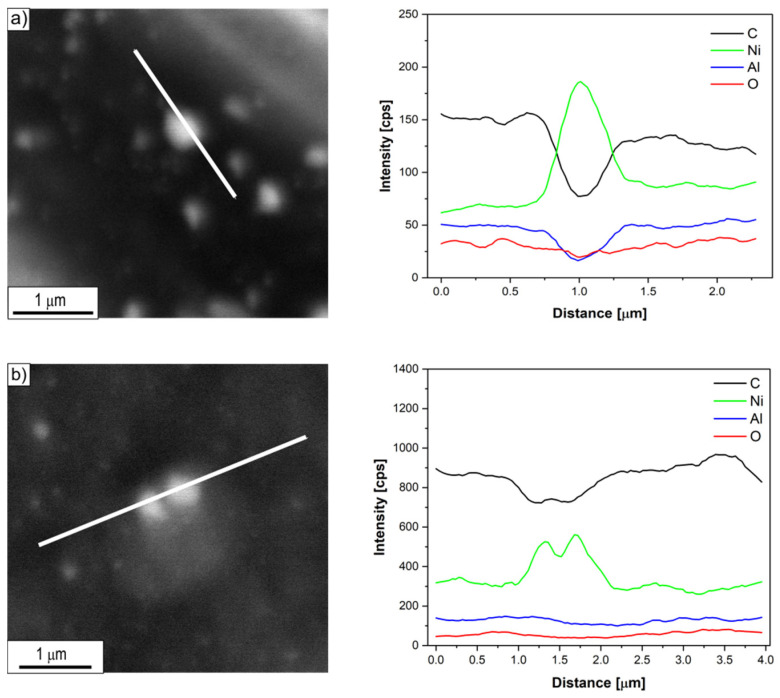
Linear EDS/SEM analysis of the CNT/CNF area with nickel-like nanoparticles formed after decomposition of ethanol with 1.1 vol.% addition of water vapour (mixture #3) at: (**a**) 500 and (**b**) 700 °C.

**Figure 8 materials-14-06086-f008:**
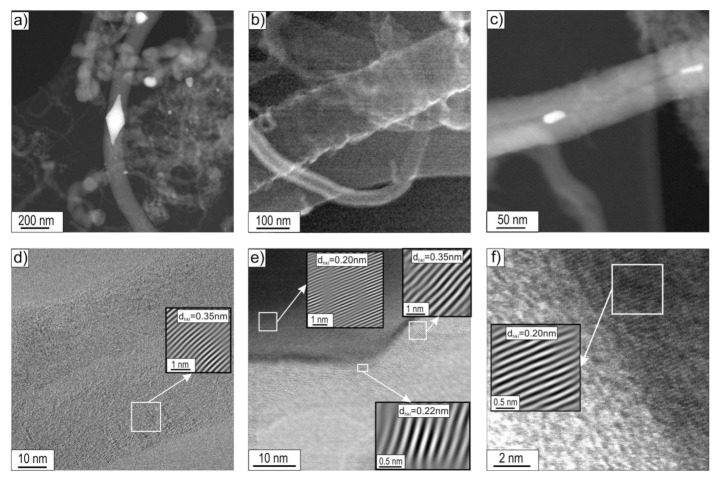
STEM morphology of deposit formed on the Ni_3_Al catalyst surface after ethanol decomposition without water vapour (i.e., mixture #1) at 500 °C: (**a**,**b**) general view and area of: (**c**) carbon nanofibre, (**d**) multiwalled carbon nanotube, (**e**) centre of nanoparticle located inside CNF, (**f**) Ni particle located inside CNF (**a**,**b**—HAADF mode, **c**–**f**—BF mode; the enlarged areas on Figure **d**–**f** were performed using IFFT).

**Figure 9 materials-14-06086-f009:**
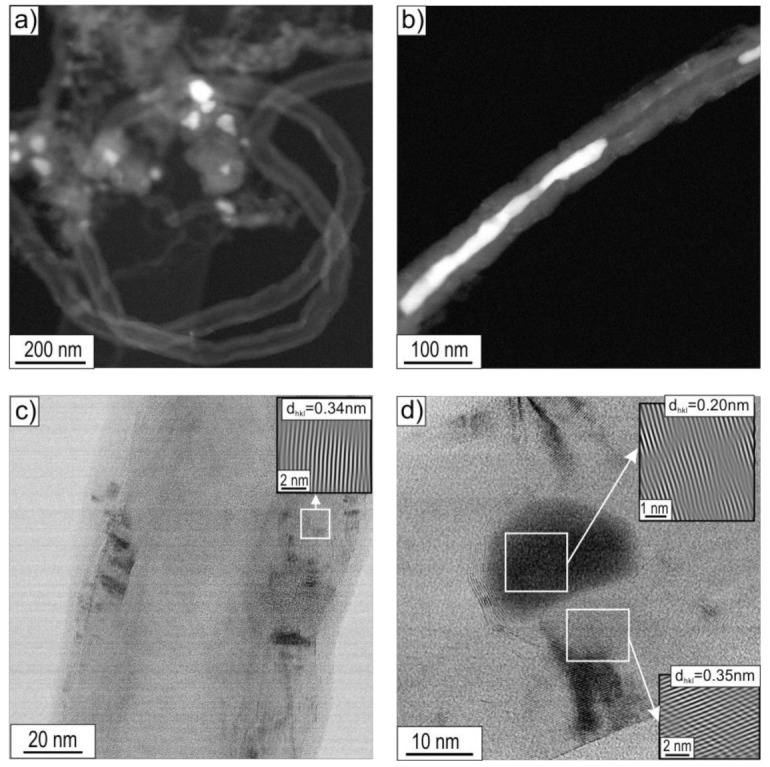
STEM morphology of deposit formed on the Ni_3_Al foils surface after decomposition of ethanol without water steam (mixture #1) at 700 °C: (**a**,**b**) general view and area of: (**c**) multiwalled carbon nanotube, (**d**) carbon nanofibre with Ni particle inside (**a**,**b**—HAADF mode; **c**,**d**—BF mode with the enlarged areas performed using IFFT).

**Figure 10 materials-14-06086-f010:**
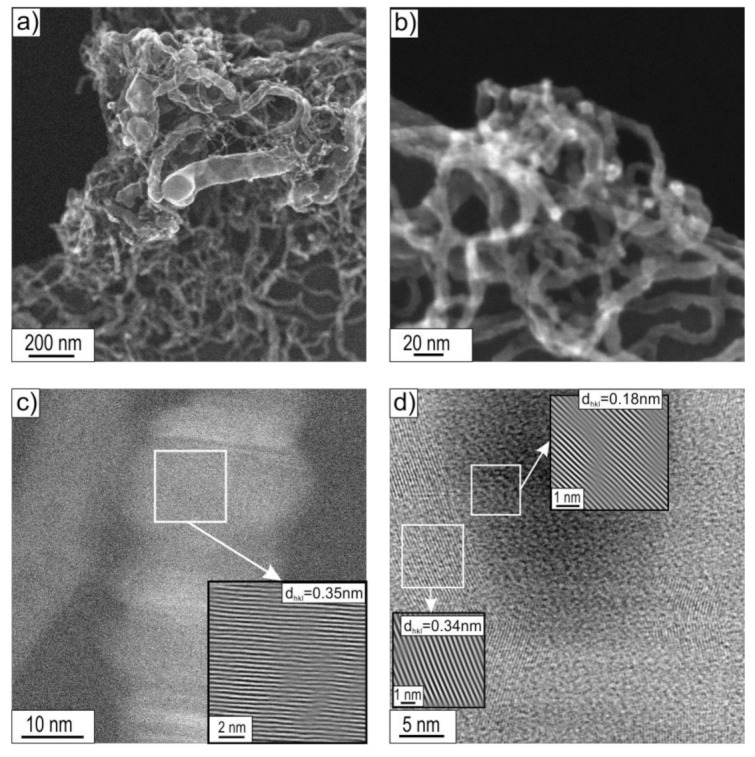
STEM morphology of deposit formed on the Ni_3_Al catalyst surface layer after decomposition of ethanol with 1.1 vol.% water vapour (mixture #3) at 500 °C: (**a,b**) general view and area of: (**c**) platelet carbon nanofibre (**d**) multiwalled carbon nanotube ended by Ni particle (**a**,**b**—SE mode; **c**—BF mode with HRTEM of selected area; **d**—BF mode with the enlarged areas performed using IFFT).

**Figure 11 materials-14-06086-f011:**
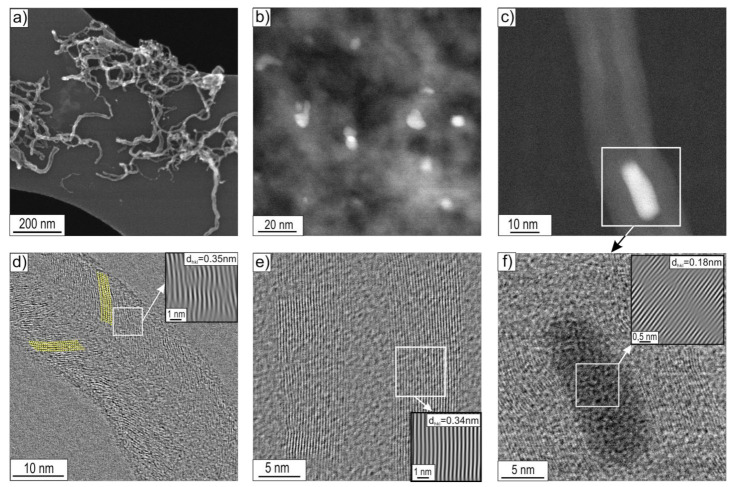
STEM morphology of deposit formed on the Ni_3_Al foils’ surface after decomposition of ethanol with 1.1 vol.% water steam (mixture #3) at 700 °C: (**a,b**) general view and area of: (**c**) multiwalled carbon nanotube with inside Ni particle, (**d**) herringbone type multiwalled carbon nanotubes, (**e**) multiwalled carbon nanotube, and (**f**) Ni particle inside MWCNT (**a**—SE mode; **b**,**c**—HAADF mode; **d**–**f**—BF mode with the enlarged areas performed using IFFT).

**Figure 12 materials-14-06086-f012:**
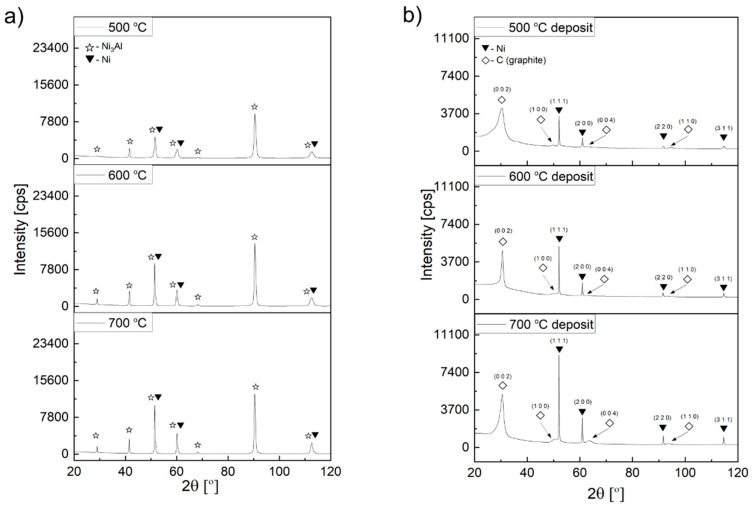
XRD profile obtained after ethanol decomposition without water vapour addition (mixture #1) from: (**a**) Ni_3_Al foils surface with carbon deposit; (**b**) deposit mechanically scrapped from the Ni_3_Al foils’ surface.

**Figure 13 materials-14-06086-f013:**
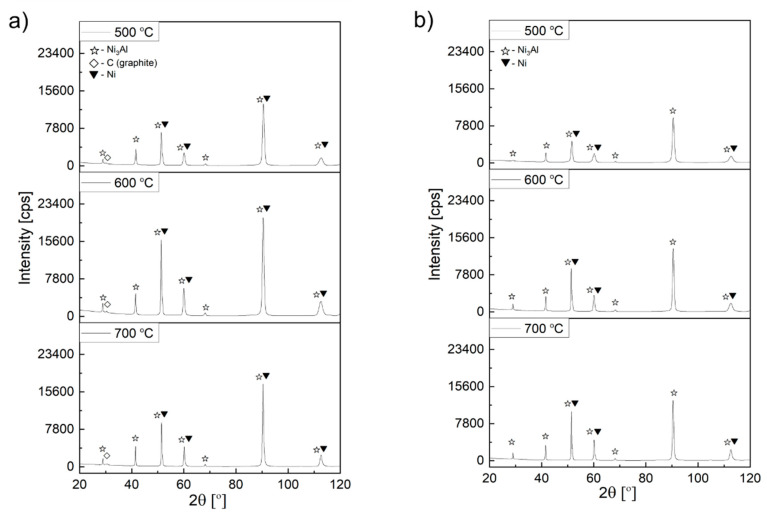
XRD profile obtained from Ni_3_Al foil surface after decomposition of: (**a**) ethanol with water vapour addition 0.17 wt.% (mixture #2) and (**b**) ethanol with 1.1 vol.% water vapour addition (mixture #3).

**Figure 14 materials-14-06086-f014:**
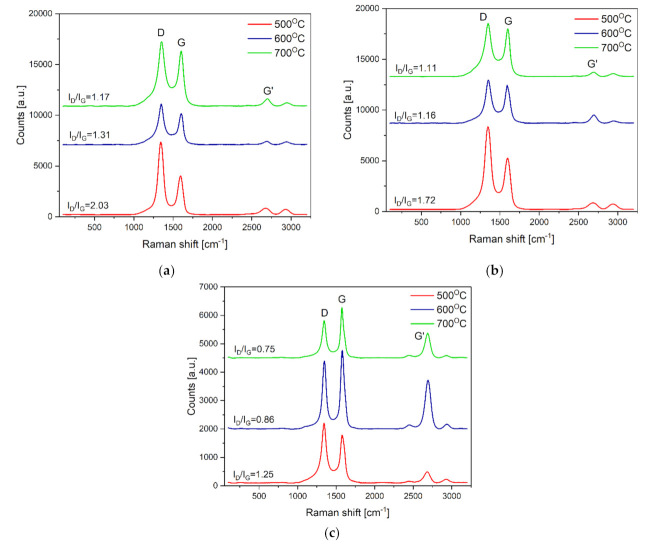
Raman spectra of Ni_3_Al foils’ surface after decomposition of ethanol: (**a**) without water vapour (mixture #1) and with water vapour addition: (**b**) 0.17 wt.% (mixture #2) and (**c**) 1.1 vol.% (mixture #3).

**Table 1 materials-14-06086-t001:** Composition of reaction mixtures used in the tests of thermocatalytic ethanol decomposition (vol.%).

Mixture	H_2_O Steam	C_2_H_5_OH (Ethanol)	Ar (Carrier Gas)
#1	-	9.70	90.30
#2	0.35	9.65	90.00
#3	1.10	9.60	89.30

## Data Availability

Not applicable.

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
