# Peer review of "Analysis of the Morphology and Structure of Carbon Deposit Formed on the Surface of Ni_3_Al Foils as a Result of Thermocatalytic Decomposition of Ethanol"

_materials, 2021, doi:10.3390/ma14206086_

Round 1

Reviewer 1 Report

This paper demonstrates the preparation of carbon materials using Ni3Al catalyst with ethanol with water. The obtained materials were characterized by SEM, STEM, XRD, and Raman spectroscopy. I consider this paper is of potential importance for the readers of Materials. My comments are listed below.

・I consider the scientific significance of this paper is not clear. The introduction section is needed to include clear scientific importance and novelty obtained from this study.

・I can not understand the motivation why water is used with ethanol. I consider only ethanol as a useful carbon feedstock for the growth of carbon nanotubes.

・How about the growth yield of carbon materials using Ni3Al catalyst?

・Why the authors used a mechanically ground treatment for Ni3Al foils? The foil does not work as the catalyst to obtain carbon materials?

Author Response

REVIEWER 1

This paper demonstrates the preparation of carbon materials using Ni3Al catalyst with ethanol with water. The obtained materials were characterized by SEM, STEM, XRD, and Raman spectroscopy. I consider this paper is of potential importance for the readers of Materials. My comments are listed below. 

  1. I consider the scientific significance of this paper is not clear. The introduction section is needed to include clear scientific importance and novelty obtained from this study.

Answer: The introduction was rewritten with a stronger indication of the scientific importance and novelty of the presented research. 

  1. I can not understand the motivation why water is used with ethanol. I consider only ethanol as a useful carbon feedstock for the growth of carbon nanotubes.

Answer: In the presented experiment, the carbon carrier for the carbon nanotubes growth was ethanol. However, water in the decomposition of hydrocarbons and their derivatives (i.e., alcohols) extends the catalyst lifetime. Due to the reduction of the presence of the solid product towards the formation of gaseous products: CO2 and H2 (i.e., steam reforming of ethanol: C2H5OH + 3H2O → 2CO2 + 6H2). For CCVD processes, where the CNTs/CNFs formation is the primary goal, a small addition of water vapour can significantly improve catalyst life and activity. The results obtained in this study also indicate that water vapour advances the received carbon nanostructures' quality.

3) How about the growth yield of carbon materials using Ni3Al catalyst? 

Answer: Both increasing the temperature and the proportion of water vapour resulted in a reduction in the presence of carbon deposit observed on the surface of Ni3Al foil. Higher temperature affects the thermodynamic parameters of the reactions occurring during ethanol decomposition. As a result, the formation of hydrogen is favored while the formation of solid products, i.e., carbon deposit, is reduced at the same time [1, 2]. On the other hand, the presence of water vapour promotes the reduction of the carbon deposit amount. It is a consequence of the transformation of carbon (present on the catalyst's surface) into gaseous products and their removal outside the reaction zone. Additional images depicting the catalyst surface are included in the paper.

The formation of carbon deposit is inherent in the thermally activated decomposition of ethanol. The presence of the Ni3Al catalyst significantly increases (compared to the reaction without the catalyst) the degree of ethanol decomposition, indirectly indicates a proportionally large increase in the mass of the carbon deposit on the Ni3Al catalyst. The obtained deposit contains carbon nanostructures of the CNT / CNF type and loosely bonded to the catalyst surface carbon subvolumes with graphite or amorphous structure. Carbon nanostructures of the CNT/CNF type play a key role in the nucleation, at the Ni3Al catalyst surface layer, of nickel nanoparticles as centers of catalytic activity. As the results show, adding water vapour to the reaction mixture with ethanol probably eliminates thermodynamically less stable forms of carbon with a structure other than CNT / CNF nanostructures. As a result, the total mass of the carbon deposit decreases (with an increase in the addition of water vapour and an increase in the decomposition temperature), but at the same time, the purity, morphological homogeneity (in terms of shape and diameter) and the perfection of the CNT / CNF nanostructure increase.

4) Why the authors used a mechanically ground treatment for Ni3Al foils? The foil does not work as the catalyst to obtain carbon materials?

Answer: As proved earlier [3-5], Ni3Al alloys show spontaneous catalytic activity without coatings or any additives. The alloy used in this work also indicates such properties.  In the case of this work, grinding was performed to mechanically remove the passive oxide nanolayer from the surface of the tested Ni3Al foil.  In our case, surface grinding of the Ni3Al foil was necessary in this case due to the earlier realization of recrystallization annealing after plastic processing of the tested intermetallic foil. Unfortunately, the protective argon atmosphere used during annealing did not fully prevent the formation of an oxide layer (mainly Al2O3) on the surface of the tested material. It was also observed in other works [3, 4].

References

  1. Wang, G.; Wang, H.; Tang, Z.; Li, W.; Bai, J., Simultaneous production of hydrogen and multi-walled carbon nanotubes by ethanol decomposition over Ni/Al2O3 catalysts. Applied Catalysis B: Environmental 2009, 88, (1-2), 142-151.
  2. Mezalira, D. Z.; Probst, L. D.; Pronier, S.; Batonneau, Y.; Batiot-Dupeyrat, C., Decomposition of ethanol over Ni/Al2O3 catalysts to produce hydrogen and carbon nanostructured materials. Journal of Molecular Catalysis A: Chemical 2011, 340, (1-2), 15-23.
  3. Behr, M. J.; Gaulding, E. A.; Mkhoyan, K. A.; Aydil, E. S., Effect of hydrogen on catalyst nanoparticles in carbon nanotube growth. Journal of Applied Physics 2010, 108, (5), 053303.
  4. Cho, W.; Schulz, M.; Shanov, V., Growth and characterization of vertically aligned centimeter long CNT arrays. Carbon 2014, 72, 264-273.
  5. Chun, D. H.; Xu, Y.; Demura, M.; Kishida, K.; Oh, M. H.; Hirano, T.; Wee, D. M., Catalytic properties of Ni 3 Al foils for methanol decomposition. Catalysis letters 2006, 106, (1), 71-75.

Reviewer 2 Report

Review report

In this paper, Jozwik et al. investigated the synthesis of CNT on Ni3Al substrates, and the effects of the experimental parameters e.g., temperature, and water on the grown materials.

This work could be interesting for some readers. However, I do not recommend the manuscript for publication in its current form.  A major revision is required before further consideration.

Here are my comments and suggestion.

  1. The introduction needs more work. In its current format, it consists of 3-4 unrelated paragraphs. The main purpose of the introduction is to tell the readers the main purpose of the work, and why this research is important or how different it is from the current-stat-of-the-art.
  2. The authors could discuss the advantages of the proposed synthesis methods compared to other methods. Since the final product of this research work is CNT, it would be better if authors start the introduction by highlighting the final product or the manufacturing process, rather than the substrate or carbon precursor (ethanol here).
  3. The last paragraph of the introduction is unclear. I found this statement “ Cylindrical carbon nanotubes made of arranged parallel to the axis graphene layers are characterized by the highest strength” . Authors need to rewrite this sentence or delete it.
  4. The last paragraph of page 2 indicated that as the content of water decreased in water/ethanol mixture, thicker carbon film was observed. It would be useful, if they could show some images or photographs from the grown films
  5. CNT/CNF was never defined in the manuscript. Before using any acronyms, they should be defined first. I suggest the use of full name (carbon nanotube/carbon nanofiber) ( for CNT/CFT)instead of abbreviations in the abstract.
  6. Line 98 and 103, what does the word “size” refer to. Is it grain size, diameter? It should be clarified through the manuscript.
  7. Why does increasing the temperature decreases the diameter of CNT/CNF? The same question for the effects of the temperature on the diameter of the grown CNT/CFT.
  8. Line 216 states “ Locally, platelet carbon nanofibers (p-CNFs) composed of graphene layers arranged perpen- 216 dicular to the fiber axis showing interplanar distances d002=0,35 nm were also found (Figure 9c) “. Figure 9c shows a HRTEM image of carbon nanotube. It is basically one-dimensional structure. There is no graphene sheet ( or any other 2D structure or layer) there. I suggest rewriting this sentence. In general terms, graphene is considered the building block of all carbon materials. However, unless you see graphene sheets or 2D structures, there is no evidence that graphene flakes exist in the grown materials.  All presented images in the paper show that the produced materials are one-dimensional, so the results should be explained and discussed accordingly.
  9. The authors mention Figure 11b multiple times in the manuscript. However, there is no label on Figure 11b. So which one is Figure 11b? More details need to be added to the Figure caption.
  10. Line 265 : “ As a consequence, it only shows the presence of graphene 265 planes corresponding to planes of type (001) (Figure 11b) “. No such a plane is indexed in any of the graphs shown in Figure 11.

Author Response

REVIEWER 2 

In this paper, Jozwik et al. investigated the synthesis of CNT on Ni3Al substrates, and the effects of the experimental parameters e.g., temperature, and water on the grown materials. 

This work could be interesting for some readers. However, I do not recommend the manuscript for publication in its current form.  A major revision is required before further consideration. 

Here are my comments and suggestion. 

  1. The introduction needs more work. In its current format, it consists of 3-4 unrelated paragraphs. The main purpose of the introduction is to tell the readers the main purpose of the work, and why this research is important or how different it is from the current-stat-of-the-art.
  2. The authors could discuss the advantages of the proposed synthesis methods compared to other methods. Since the final product of this research work is CNT, it would be better if authors start the introduction by highlighting the final product or the manufacturing process, rather than the substrate or carbon precursor (ethanol here).
  3. The last paragraph of the introduction is unclear. I found this statement "Cylindrical carbon nanotubes made of arranged parallel to the axis graphene layers are characterized by the highest strength". Authors need to rewrite this sentence or delete it.

Answer to comments 1-3: The introduction has been rewritten with a stronger indication of the scientific significance and novelty of the research presented. A more detailed presentation of the advantages of the proposed synthesis method over others can be found in the introduction. The entire chapter has been rewritten, emphasizing important aspects in the context of the analysis of the obtained experimental data.

  1. The last paragraph of page 2 indicated that as the content of water decreased in water/ethanol mixture, thicker carbon film was observed. It would be useful, if they could show some images or photographs from the grown films.

Answer: As suggested by the Reviewer, appropriate micrographs were attached to the manuscript (complex Fig. 1).

  1. CNT/CNF was never defined in the manuscript. Before using any acronyms, they should be defined first. I suggest the use of full name (carbon nanotube/carbon nanofiber) ( for CNT/CFT)instead of abbreviations in the abstract.

Answer: The acronyms CNT/CNF are commonly used in the literature; therefore, the authors did not explain their sense. But, formally, the full name of each term should be used in the beginning. According to the Reviewer's suggestion, appropriate changes were made.  

  1. Line 98 and 103, what does the word "size" refer to. Is it grain size, diameter? It should be clarified through the manuscript.

Answer: The word "size" refers to the diameter of the carbon nanostructures. This was changed in the manuscript.

  1. Why does increasing the temperature decreases the diameter of CNT/CNF? The same question for the effects of the temperature on the diameter of the grown CNT/CFT.

Answer: In the second part of this question, the Reviewer probably asks about the influence of the addition of water vapour, not temperature (for the second time), on the effect of ethanol's thermal catalysis of ethanol?

The decomposition temperature is an important parameter influencing all the phenomena accompanying the analyzed process, including the conditions and course of the chemical compound decomposition reaction and changes in the catalyst surface layer. It is also a factor that significantly impacts the formation mechanism, morphology, and substructure of the deposit with the participation of carbon nanostructures.

First, higher temperature influences the thermodynamic parameters of the reactions taking place during the decomposition of ethanol. As a result, the release of hydrogen is favoured [1, 2], while increasing the amount of hydrogen in the reaction space leads to a decrease in the average diameter of CNT/CNF [5, 6].

According to the research [7-9], the temperature of the decomposition process also significantly influences the activation of the Ni3Al catalyst in the form of a foil or powder. As a result of the selective oxidation/hydroxylation of aluminium in the Ni3Al structure, catalytically active Ni nanoparticles are formed in the surface layer of the catalyst. Higher temperature intensifies this activation process, resulting in a greater number of nuclei and a smaller size of Ni nanoparticles.

During the decomposition of ethanol, these nanoparticles are separated from the catalyst surface layer. They are "transported" on the formed CNT/CNF towards the thermocatalytic reaction zone (at the border with the reaction mixture). Such a mechanism of "separation" of nickel nanoparticles from each other prevents their agglomeration and sintering (with an inevitable increase in the size of Ni nanoparticles), which ultimately results in the reduction of CNT/CNF diameter, closely related to the size of the active Ni nanocenters.

The shape of the catalytically active Ni particles also changes with increasing temperature. Nickel nanoparticles gradually elongate (to the shape of a cone), which favours the transformation of carbon atom layers (graphene-like layers) into tubular/fibrous carbon nanostructures. The increase in decomposition temperature also results in an increased presence of elongated Ni nanoparticles encapsulated in CNT / CNF along the length of these carbon nanostructures.

Regarding the influence of water vapour in the reaction mixture - in the presented experiment, the carbon carrier for the growth of carbon nanotubes was, of course, ethanol. However, water in the decomposition of hydrocarbons and their derivatives (i.e., alcohols) extends the life of the catalyst. This is due to the reduction of the presence of solid decomposition products towards the formation of gaseous products: CO2 and H2 (i.e., steam reforming of ethanol: C2H5OH + 3H2O → 2CO2 + 6H2). For CCVD processes where CNT/CNF formation is the primary goal, a small addition of water vapour can significantly improve catalyst life and activity. The results obtained in this study also indicate that water vapour significantly improves the morphology and substructure of the obtained carbon nanostructures.

As the results show, adding water vapour to the reaction mixture with ethanol probably eliminates thermodynamically less stable forms of carbon with a structure other than CNT/CNF nanostructures. As a result, the total mass of the carbon deposit decreases (with an increase in the addition of water vapour and an increase in the decomposition temperature), but at the same time, the purity, morphological homogeneity (in terms of shape and diameter) and the perfection of the CNT/CNF nanostructure increase.

  1. Line 216 states "Locally, platelet carbon nanofibers (p-CNFs) composed of graphene layers arranged perpendicular to the fiber axis showing interplanar distances d002=0,35 nm were also found (Figure 9c) ". Figure 9c shows a HRTEM image of carbon nanotube. It is basically one-dimensional structure. There is no graphene sheet ( or any other 2D structure or layer) there. I suggest rewriting this sentence. In general terms, graphene is considered the building block of all carbon materials. However, unless you see graphene sheets or 2D structures, there is no evidence that graphene flakes exist in the grown materials.  All presented images in the paper show that the produced materials are one-dimensional, so the results should be explained and discussed accordingly.

Answer: In describing the HRTEM images presented in this article, the authors used an approach commonly found in the literature. In many (mostly) scientific works, obtained and tested nanotubes/carbon fibres are treated as nanoforms made of graphene layers [10-16], not as theoretical one-dimensional structures. It should be noted that the diameter of carbon nanoforms observed in the conducted research is expressed by the value of tens or even hundreds and not individual nanometers (as in the case of one-dimensional structures). The presence of the basic crystallographic plane (002) in the graphite structure in the X-ray diffraction patterns of the investigated carbon deposit (although assessed globally) also provided some clue to this statement (Fig. 12b).

At the same time, the authors agree with the Reviewer that the term "graphene planes," commonly used in literature and this work, is not entirely correct and sufficiently proven. Consequently, as suggested by the Reviewer, appropriate changes were made to the text, i.e. "graphene layers" were replaced (in the relevant parts of the text) with "parallel layers of carbon atoms", "graphene-like layers" or "layers of carbon atoms, identical to the crystallographic plane (002) of graphite".

  1. The authors mention Figure 11b multiple times in the manuscript. However, there is no label on Figure 11b. So which one is Figure 11b? More details need to be added to the Figure caption.

Answer: The proper changes were made.

  1. Line 265 : "As a consequence, it only shows the presence of graphene 265 planes corresponding to planes of type (001) (Figure 11b) ". No such a plane is indexed in any of the graphs shown in Figure 11.

Answer: The authors agree with the Reviewer - none of the diffraction patterns in Fig. 11b show (as it would not be physically possible) a reflex from the (001) plane in the graphite structure. The sentence in line 265 of the manuscript should sound: "As a consequence, it only shows the presence of reflex from parallel layers of carbon atoms corresponding to planes of type (002) in graphite structure (Figure 12b)".

References

  1. Wang, G.; Wang, H.; Tang, Z.; Li, W.; Bai, J., Simultaneous production of hydrogen and multi-walled carbon nanotubes by ethanol decomposition over Ni/Al2O3 catalysts. Applied Catalysis B: Environmental 2009, 88, (1-2), 142-151.
  2. Mezalira, D. Z.; Probst, L. D.; Pronier, S.; Batonneau, Y.; Batiot-Dupeyrat, C., Decomposition of ethanol over Ni/Al2O3 catalysts to produce hydrogen and carbon nanostructured materials. Journal of Molecular Catalysis A: Chemical 2011, 340, (1-2), 15-23.
  3. Behr, M. J.; Gaulding, E. A.; Mkhoyan, K. A.; Aydil, E. S., Effect of hydrogen on catalyst nanoparticles in carbon nanotube growth. Journal of Applied Physics 2010, 108, (5), 053303.
  4. Cho, W.; Schulz, M.; Shanov, V., Growth and characterization of vertically aligned centimeter long CNT arrays. Carbon 2014, 72, 264-273.
  5. Chun, D. H.; Xu, Y.; Demura, M.; Kishida, K.; Oh, M. H.; Hirano, T.; Wee, D. M., Catalytic properties of Ni 3 Al foils for methanol decomposition. Catalysis letters 2006, 106, (1), 71-75.
  6. Chun, D. H.; Xu, Y.; Demura, M.; Kishida, K.; Wee, D. M.; Hirano, T., Spontaneous catalytic activation of Ni3Al thin foils in methanol decomposition. Journal of Catalysis 2006, 243, (1), 99-107.
  7. Hirano, T.; Xu, Y.; Demura, M. In Catalytic properties of Ni3Al foils for hydrogen production, Advanced Materials Research, 2011; Trans Tech Publ: pp 130-133.
  8. Martin-Gullon, I.; Vera, J.; Conesa, J. A.; González, J. L.; Merino, C., Differences between carbon nanofibers produced using Fe and Ni catalysts in a floating catalyst reactor. Carbon 2006, 44, (8), 1572-1580.
  9. Hu, J.; Odom, T. W.; Lieber, C. M., Chemistry and physics in one dimension: synthesis and properties of nanowires and nanotubes. Accounts of chemical research 1999, 32, (5), 435-445.
  10. Serp, P.; Corrias, M.; Kalck, P., Carbon nanotubes and nanofibers in catalysis. Applied Catalysis A: General 2003, 253, (2), 337-358.
  11. De Volder, M. F.; Tawfick, S. H.; Baughman, R. H.; Hart, A. J., Carbon nanotubes: present and future commercial applications. science 2013, 339, (6119), 535-539.
  12. Monthioux, M.; Serp, P.; Caussat, B.; Flahaut, E.; Razafinimanana, M.; Valensi, F.; Laurent, C.; Peigney, A.; Mesguich, D.; Weibel, A., Carbon nanotubes. In Springer Handbook of Nanotechnology, Springer: 2017; pp 193-247.
  13. Matsumoto, K., Frontiers of graphene and carbon nanotubes: devices and applications. Springer: 2015.
  14. Endo, M.; Takeuchi, K.; Hiraoka, T.; Furuta, T.; Kasai, T.; Sun, X.; Kiang, C.-H.; Dresselhaus, M., Stacking nature of graphene layers in carbon nanotubes and nanofibres. Journal of Physics and Chemistry of Solids 1997, 58, (11), 1707-1712.
  15. Maniecki, T.; Shtyka, O.; Mierczynski, P.; Ciesielski, R.; Czylkowska, A.; Leyko, J.; Mitukiewicz, G.; Dubkov, S.; Gromov, D., Carbon nanotubes: properties, synthesis, and application. Fibre Chemistry 2018, 50, (4), 297-300.
  16. Belin, T.; Epron, F., Characterization methods of carbon nanotubes: a review. Materials Science and Engineering: B 2005, 119, (2), 105-118.

Round 2

Reviewer 1 Report

The manuscript has been improved and answered along with the referees' comments. I consider this paper has appropriate importance; therefore, I recommend the publication of this paper.

Reviewer 2 Report

I recommend the publication of the revised manuscript.